# Reduction of stillbirth rate in refugee and migrant populations living on the Thailand Myanmar border: A retrospective study 1986–2023

Taco Jan Prins[1,2,3]*, Gwen van der Schaaf[4], Aung Myat Min[4], Mary Ellen Gilder[4,5], Nay Win Tun[4], May Mon Mon Theint[4], Jacher Wiladphaingern[4], Eh Moo[4], Kerryn A. Moore[6], Jelle Stekelenburg[7], Chaisiri Angkurawaranon[1,2], Marcus J Rijken[4,8], Michele van Vugt[3,4], François Nosten[4,5], Rose McGready[4,5]

**1** Department of Family Medicine, Faculty of Medicine, Chiang Mai University, Chiang Mai, Thailand, **2** Global Health and Chronic Conditions Research Group, Chiang Mai University, Chiang Mai, Thailand, **3** Amsterdam University Medical Centres, Department of Internal Medicine and Infectious diseases, and Research groups: APH, GH and AII&I, Amsterdam UMC, Amsterdam, The Netherlands, **4** Shoklo Malaria Research Unit, Mahidol–Oxford Tropical Medicine Research Unit, Faculty of Tropical Medicine, Mahidol University, Mae Ramat, Thailand, **5** Centre for Tropical Medicine and Global Health, Nuffield Department of Medicine, University of Oxford, Oxford, United Kingdom, **6** Centre for Epidemiology and Biostatistics, Melbourne School of Population and Global Health, The University of Melbourne, Melbourne, Australia, **7** Department of Health Science, Global Health, University Medical Centre Groningen, Groningen, Netherlands, **8** Julius Global Health, Julius Centre for Health Sciences and Primary Care, University Medical Centre Utrecht, Utrecht University, The Netherlands

* tacojan@shoklo-unit.com

## Abstract

Understanding the local causes of stillbirth is essential to providing safe care in pregnancy and birth. This study describes the rate and causes of stillbirth and identifies factors associated with stillbirth in the migrant and refugee population residing in border regions between Thailand and Myanmar. A retrospective review of medical records of all singleton pregnancies delivered or followed at antenatal clinics of the Shoklo Malaria Research Unit from 1986 to 2023, with a known outcome of pregnancy and estimated gestational age of 28 weeks and more. Multivariable logistic regression was performed to compare the factors between livebirths and stillbirths. During the 37 years period there were 65,101 singleton births including 721 stillbirths. The stillbirth rate decreased to a third of the initial rate from 26 per 1000 (49/1,904) [95% Confidence interval (CI): 19–34] in 1986–1990–8 per 1000 (66/7,790) [95% CI: 7–11] in 2020–2023. In 1986, 80% of births took place at home with the proportion declining significantly over time settling to 10–15% from 2012. Out of 721 stillbirths, 574 (79.5%) were classifiable. 26.0% (149/574) of those were designated as intrapartum and 74.0% (425/574) as antepartum stillbirth. Causal classification was possible for 67.7% (488/721) of stillbirths, with the top 3 causes being antepartum haemorrhage 26.0% (127/488), congenital abnormality 12.9% 63/488); and maternal infection 12.3% (60/488). It is possible to decrease the stillbirth rate in refugee and

**Data availability statement:** Data cannot be shared publicly because this is a population of undocumented refugees and migrants and we do not have their permission to share their data. Data are available from the Mahidol-Oxford Research Unit Institutional data access committee upon reasonable request from researchers who meet the criteria for access to confidential data. Enquiries should be directed through the Bioethics and Engagement team coordinates Mahidol Oxford Research Unit's Data Access Committee (datasharing@tropmedres.ac).

**Funding:** This study was partially supported by Chiang Mai University (https://www.cmu.ac.th) in the form of salary received by TJP and CA. The specific roles of these authors are articulated in the 'author contributions' section. The Shoklo Malaria Research Unit is supported in part by the Wellcome-Trust Major Overseas Programme in Southeast Asia (https://doi.org/10.35802/220211) in the form of a grant (220211). This research was funded in part, by the Wellcome Trust [315982/Z/24/Z]. For the purpose of Open Access, the author has applied a CC BY public copyright licence to any Author Accepted Manuscript version arising from this submission. No additional external funding was received for this study. The funders had no role in study design, data collection and analysis, decision to publish, or preparation of the manuscript.

**Competing interests:** The authors have declared that no competing interests exist.

**Abbreviations:** OR: Odds Ratio, PPH: postpartum haemorrhage, ANC: antenatal care, SMRU: Shoklo Malaria Research Unit, TBA: traditional birth attendant, EGA: estimated gestational age, APH: antepartum haemorrhage, SGA: Small for gestational age, CI: confidence interval, EmOC: Emergency obstetric Course, ALSO: Advanced Life Support in Obstetrics, LGA: Large for gestational age, DAG: Direct Acyclic Graph, IQR: interquartile range.

migrant populations despite intermittent conflict. Improving access to health clinics and skilled attendance at birth has the potential to reduce preventable stillbirths.

## Introduction

In 2010, the World Health Organization initiated the Every Woman Every Child movement and in 2014 the Every Newborn Action Plan to reduce the preventable deaths for all infants, mothers and newborns, including stillbirth and early neonatal death [1,2]. The global stillbirth rate decreased from 21.3 per 1000 births from 28 weeks gestation in 2000 to 13.1 per 1000 births in 2021 [3]. The majority of these stillbirths are in low resource settings such as sub Saharan-Africa and Southern Asia, with a rate of 21.0 per 1000 births and 16.7 per 1000 births respectively [3,4] while high resource settings such as Europe record 3.0 stillbirths per 1000 births [3].

The frequency and causes of stillbirth vary between countries and settings, with a higher percentage of stillbirths with an unknown cause in low resource settings (41%) compared to high resource settings (32%) [5]. Internal audits to review and classify stillbirths are important to identify causes and improve perinatal outcomes [6]. However, as registration of births can be absent or insufficient in most low-resource settings, many stillbirths occur without documentation [1,7]. Understanding the local causes of stillbirth is essential to providing safe care in pregnancy and during birth. In theory, most stillbirths are preventable if care is accessible and of quality across the continuum of care, including timely and appropriate antenatal, intrapartum and referral care [3,4,8]. Marginalised populations often have more difficulty accessing these types of care [9,10]. For refugee and migrant women living on the border of Myanmar and Thailand, the complicated political, geographical and socio-economic situations frequently impedes access to quality care [11,12]. Apart from focused analysis on the association between *Plasmodium falciparum* and *P. vivax* malaria in pregnancy and stillbirth, stillbirths and its specific causes for this marginalised population in this fragile environment have not been examined [13].

The objective of this study is to describe the rates and causes of stillbirth, if known, and identify the maternal and fetal factors during pregnancy, birth and the post-partum period associated with stillbirth in the migrant and refugee population residing on the border between Thailand and Myanmar. The period of data analysis is from 1986 to 2023 - a time marked by intermittent conflicts and displacements.

## Methods

### Ethical approval

Ethical approval for retrospective analysis of hospital archives at the Shoklo Malaria Research Unit (SMRU) was provided by the Oxford University Ethics Committee (OXTREC: 28–09), by the Research Ethics Committee, Faculty of Medicine, Chiang Mai University (0565/2023) and the local community advisory board in Mae-Sot, Thailand (TCAB-4/1/2015).

## Design

Medical records from the SMRU Refugee and Migrant Pregnancy study including all singleton pregnancies where women followed antenatal care (ANC) and/or gave birth at the clinics of SMRU, were reviewed retrospectively [14]. Multiple pregnancies were not included since a recent study in this community reported a low rate (1% of all births) and consistent with international data, an increased risk of stillbirth in these pregnancies [15]. The STROBE guideline for observational studies was followed for reporting [16]. The clinical details were extracted from medical records which were initially paper based and became electronic in from 2008, with one clinic at a time making the transition as direct mentorship was required. Paper-based records prior to 2008 were extracted manually to an electronic database so that ANC and birth outcome data could be retrieved electronically. Information from records such as the paper-based partogram or the hospital referral forms was retrieved from archives stored at SMRU for all stillbirths and if needed for livebirths when required for clarification. The dataset for the purposes of this research was first accessed and extracted on 08/01/2024, updating and clarification was completed by 30/01/2025. All identifiable information of the women was removed or anonymised.

## Setting

Since 1986, the SMRU has built up a system of clinics providing ANC and perinatal care on both sides the Thai-Myanmar border in Tak Province Thailand. The initial focus besides perinatal care was screening and treating of malaria, which causes high morbidity among this population. Services were developed first for refugee women in camps in Thailand (between 1986 and 2016) followed by services for migrant women (from either side of the border). Migrant clinics provided ANC starting from 1997 and from 2007 perinatal care. After closure of SMRU's last refugee clinic in 2016 and later with the COVID-19 pandemic, clinics on the Myanmar side of the border were opened in 2017 and 2020 to serve the marginalized and displaced populations – who can be labelled as internally displaced [17]. The migrant population in Thailand and internal displaced population inside Myanmar fluctuate, for example decreasing with the semi-democratic rule in the late 2010s and increasing with the armed conflict of the 2021 coup d'etat. The population and clinics in the period of time have been described in more detail, including maps, in the SMRU Refugee and Migrant Pregnancy Cohort [14]. The populations consist mostly of Burmese and Karen, and less frequently other ethnic groups as Shan or Mon. The impact of various barriers: language, health insurance, COVID-19 pandemic and geography on access to ANC and pregnancy outcomes in this politically complex border region has been described earlier [11,17,18]. The ANC and perinatal care at SMRU clinics are mainly midwife led. Midwives are either SMRU-trained (attending a 15-month training course, described previously [19]), or (since the 2021 military coup) displaced trained midwives from Myanmar. Doctors trained in obstetrics support the midwifes and ANC teams at all sites. Midwives have regular training and have participated in emergency obstetric care (EmOC) courses such as the Advanced Life Support in Obstetrics (ALSO) course [20]. The WHO partograph was introduced in 1994 and included recording of fetal distress, meconium-stained amniotic fluid, tachysystole (excessive uterine activity), uterine rupture, abnormal maternal blood pressure during labour, mode of birth, Apgar score, neonatal resuscitation and postpartum haemorrhage (PPH) [20,21]. Births completed at SMRU facilities are referred to as clinic births. Complicated births requiring caesarean section are referred to a local Thai or Myanmar hospital (15 minute to one hour drive) with an emergency car kept on standby for 24 hours a day. These births are referred to as hospital births. SMRU has no access to the Thai or Myanmar Government hospitals and collect the data of referred patients, retrospectively after discharge.

## Participants

All women with singleton pregnancies who followed ANC and/or gave birth at SMRU clinics were included. If the outcome of the pregnancy was not known in this mobile population, or estimated gestational age (EGA) was not known, the record was excluded. In this resource limited setting active management is provided from an EGA of 28 weeks and all pregnancy

outcomes with an EGA below 28 weeks were excluded. Previous research already described the low percentage of survival of births before 28 weeks and the overlap with unidentified miscarriage [22,12]. Missing data per variable extracted is presented in a table (Table A in S1 Appendix).

## Variables

The distinction between migrant and refugee was made on basis of where the woman was registered for ANC: refugee camp or migrant population. Anaemia was classified as haematocrit less than 30% measured during pregnancy. Hypertensive disease in pregnancy was defined as a blood pressure above or equal to 140/90 mmHg measured twice at least 6 hours apart. Pre-eclampsia was described as hypertension accompanied by the presence of protein in the urine, tested with urine dipstick and/or complaints such as changes in vision, headache, oedema, or vomiting. Preterm birth was described as birth with an EGA < 37 weeks, and further categorized with ≥28 and <32 weeks (very preterm) and ≥32 and <37 (moderate and late preterm). EGA was determined using ultrasound (2001–2023) [23]. Prior to 2001 Dubowitz Assessment of gestation after birth, a population specific model based on ultrasound for women with multiple symphysis fundal height measurements [24] was done and if not possible last menstrual period or by the woman's reported self-estimate at first ANC. After 2001 if the women arrived for first ANC after an EGA of 28 + 0 weeks, Dubowitz assessment of gestation after birth was done. Antepartum haemorrhage (APH) was defined as antepartum blood loss (EGA ≥ 28 weeks). Small for gestational age (SGA) and large for gestational age (LGA) were defined as <10th and >90th percentile of birthweight for gestational age and sex using the international standards of the INTERBIO-21st Project [25]. Congenital abnormalities were assessed through systematic physical exam, routinely done since 1998, and defined as any observed abnormality. BMI was determined in all women at first ANC, but analysis was limited to women who attended in the first trimester, as a proxy for pre-pregnancy BMI. Infection with malaria during pregnancy was considered present if positive malaria smear was detected at any ANC visits. The species of malaria by microscopy were *P. vivax* or *P. falciparum or both (a mixed infection)*. If there were both *P. falciparum* and *P. vivax* infection during infection, this was counted as *P. falciparum*, as this is the most severe. Stillbirth rate was defined as number of stillbirths per 1,000 total births from EGA ≥ 28 weeks. Stillbirths were categorized into antepartum, intrapartum or unknown. Antepartum death was defined as all fetal deaths occurring before the onset of the active phase of labour, the active phase being cervical dilation of more than 3 centimetres accompanied by regular contractions. Intrapartum death was defined as all fetal death occurring between the onset of the active phase of labour and birth. If there was no partogram present, individual clinic records were reviewed for information on fetal heartbeat including phase of labour, the last time fetal movement was felt or if maceration at birth was seen. APH was assumed as antepartum stillbirth if no partogram was present. Due to unknown timing of death (e.g., antepartum or intrapartum) most stillbirths at home had an unknown timing of the stillbirth, unless there was a strong indication for the timing of death, for antepartum (e.g., severe growth restriction or maceration) or intrapartum (arrived in the clinic with head entrapment at breech delivery, or prolonged entrapment was well described by patient/patient relative/traditional birth attendant and noted in the record).

The suspected clinical causes of stillbirth were classified through a modified version of the ReCoDe [26]. Antepartum haemorrhage: placenta abruption or placenta praevia, Congenital abnormality, Placenta insufficiency (SGA/Intra uterine growth restriction/oligohydramnios/histologic changes in placenta), Maternal Infection, Maternal illness, Hypertensive diseases in pregnancy (Hypertension/Pre-Eclampsia, Eclampsia), Cord (Cord prolapse, cord constriction), Obstructive or prolonged labour (including malpresentations and shoulder dystocia), Uterine Rupture, Trauma (by fall or pushing on the abdomen by traditional birth attendant), Acute intrapartum moment, Other, or Unknown. The details per category are specified in Table B in S1 Appendix. Additionally, the International Classification of Disease 10 Perinatal-Mortality (ICD-PM) was used for classification [6]. The classification was conducted by two individuals separately (GvdS and TJP) to increase its reliability and a third (RM) resolved discrepancies by consensus. If there was insufficient information, it was classified as unknown.

## Statistical analysis

Data were analysed using R (R Core Team (2024) (version 4.4.2) for Windows. Continuous data were described using the mean, median, standard deviation, interquartile range [IQR] and range. Binary and categorical data were summarized using frequency and proportions. Multivariable logistic regression was performed to compare the factors between livebirths and stillbirths. A Directed Acyclic Graph (DAG) (S1 Fig) was made including the associated factors: literacy, anaemia, APH, SGA, hypertensive diseases, HIV, Syphilis, malaria, age, parity, smoking and preterm labour, and multivariable analysis was done for those variables based on the DAG. All models were adjusted for year of birth to account for temporal trends, not shown in the DAG for clarity. Preterm birth was not taken into the multivariable analysis as it can be both a consequence of stillbirth and influenced by factors that cause stillbirth. Age and parity had a moderate correlation (Pearsons/Spearman = 0.71). However, variance inflation factors for both variables were low (2.08), indicating no concern for multicollinearity. Therefore, both variables were included in the model.

## Results

During the period from 1986 till 2023 there were 65,101 singleton births with an EGA of 28 weeks or more and a known outcome (Fig 1). There was a total of 721 stillbirths (overall stillbirth rate of 11 per 1000 [721/65,101; 95% CI: 11–12]). The stillbirth rate decreased to a third of the initial rate from 26 per 1000 (49/1,904) [95% Confidence interval (CI): 19–34] in 1986–1990–8 per 1000 (66/7,790) [95% CI: 7–11] in 2020–2023. This decrease was largely achieved before the year 2000. A slight rise was seen during the addition of migrant antenatal care services, which added a previously unserved high-risk group without reliable access to safe delivery care, and fell after delivery rooms were opened for migrant women in 2007 settling at around 5–10 per 1000 (Fig 2). In the same time period, the amount of homebirth and *P. falciparum* infections declined. Smoking also declined significantly from 40.1% (780/1,946) in 1998–1999 to 3.9% (186/4,929) in 2022–2023 (S2 Fig). Missing data per variable extracted is available (Table A in S1 Appendix).

### Causes of stillbirth

Of the 721 stillbirths, 67.7% (488/721) could be attributed a cause (Table 1). Of the 233 that could not be classified (32.3% [233/721]), half (49.8% [116/233]) took place outside the clinic/hospital. The most prevalent causes of stillbirth were: APH (26.0% [127/488]) of which 108 were placenta abruptio and 19 placenta praevia; congenital abnormality (12.9% [63/488]); and maternal infection (12.3% [60/488]), including 15 cases of suspected chorioamnionitis and 18 cases of malaria at term. Obstructed or prolonged labour caused 48 stillbirths (9.8% [48/488]), most commonly due to a difficult breech birth (47.9% [23/48]). Of the 63 stillbirths caused by congenital abnormality, three pregnancies were medically terminated, two of them because of fetal anencephaly and one with severe fetal hydrocephalus. A special attention needs to be given for

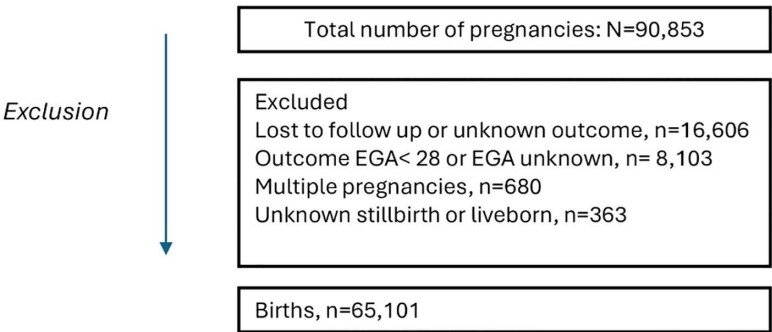

**Fig 1. Flowchart exclusion.**

## Trends in maternal and perinatal indicators over time

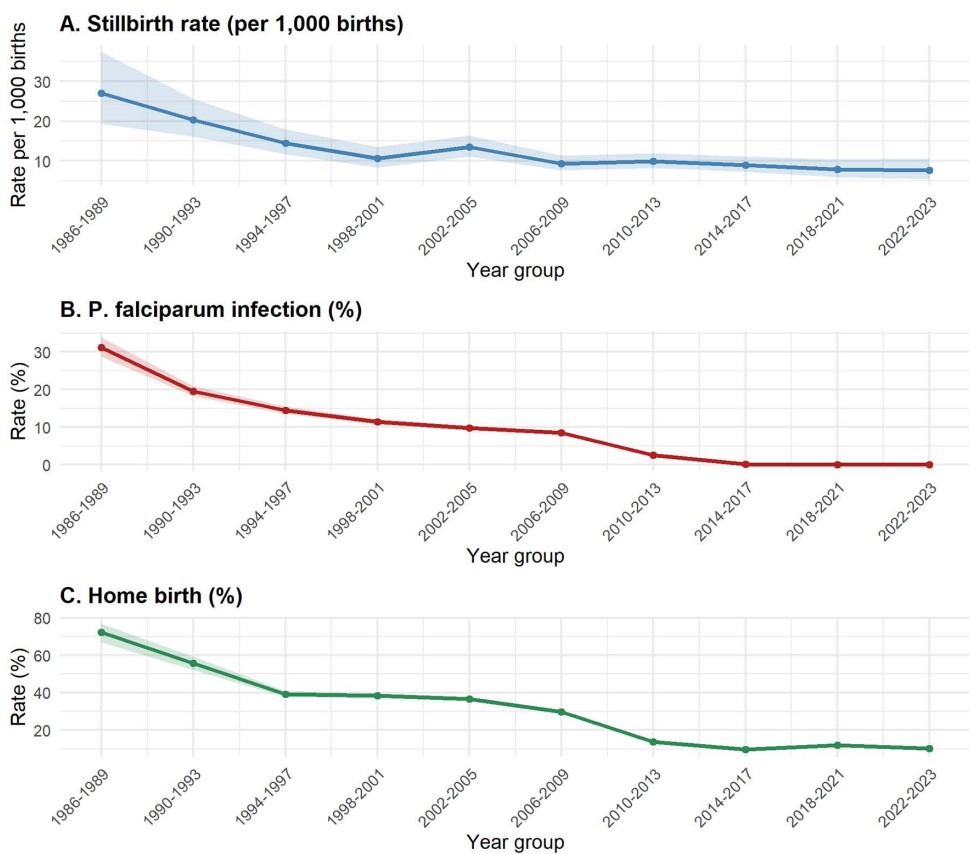

**Fig 2. Trend in maternal and perinatal indicators over time.**

**Table 1. Causes of stillbirth in ante and intrapartum classified cases.**

| Causes | N (%) (total n = 721) |
|---|---|
| Unknown | 233 (32.3) |
| **KNOWN CAUSES** | **N = 488** |
| Antepartum haemorrhage | 127 (26.0) |
| Congenital abnormality | 63 (12.9) |
| Maternal infection | 60 (12.3) |
| Placenta insufficiency | 55 (11.3) |
| Obstructed or prolonged labour | 48 (9.8) |
| Hypertensive diseases | 38 (7.8) |
| Cord | 30 (6.1) |
| Trauma | 24 (4.9) |
| Uterine rupture | 13 (2.7) |
| Maternal illness | 12 (2.5) |
| Acute intrapartum moment | 11 (2.3) |
| Other | 7 (1.4) |

the 24 cases of trauma: in (2.5%, 12/488) cases women reported the traditional birth attendant (TBA), frequently along with other attendants, had given a strong massage, or applied strong pressure on the uterus which caused bruising or loss of fetal movement. In addition, of the 13 cases of uterine rupture, (69.2%, 9/13) of cases occurred before arrival to the clinic with a clear history of TBA and/or family members pushing strongly on the uterus believing it will facilitate birth. The stillbirths were additionally classified using the WHO ICD 10-PM (Table C in S1 Appendix). Of all the stillbirths that could be classified, 26.0% (149/574) were designated as intrapartum and 74.0% (425/574) as antepartum stillbirths.

## Characteristics and associated factors

Table 2 shows the characteristics of the women with livebirths and stillbirths. APH had the greatest association with stillbirth (adjOR 24.28 (18.28-31.94)), followed by congenital abnormality (adjOR 11.17 (95% CI 8.79-14.05)) HIV (adjOR 5.06 (95% CI 1.77-11.39) and small for gestational age (adjOR 3.01 (95% CI 2.45-3.70)) (Table 3). Of the 5 stillbirths of pregnant women with HIV, 2 had a *P. falciparum* infection during pregnancy and another tested positive also for syphilis. The percentage of APH decreased over time: 1.5% (86/5,608) [95% CI: 1.2-1.9] in 2000–2003 to 0.6% (47/7,779) [95% CI: 0.4-0.8] in 2020–2023. Of the malaria species, *P. falciparum* (including symptomatic and asymptomatic infections) was associated with stillbirth (adjOR 1.62 (95% CI 1.16-2.20), but *P. vivax* (including symptomatic and asymptomatic infections) was not (adjOR 0.77 (95% CI 0.49-1.14)).

## Antepartum and intrapartum stillbirth

The proportion of unknown causes was higher in the antepartum 27.5% (117/425) compared to the intrapartum 4.1% (6/149) group. Among the known causes of antepartum stillbirth the following causes were most common: APH 36.0% (111/308), maternal infection 15.6% (48/308) and placenta insufficiency 13.0% (40/308) (Table 4).

For intrapartum stillbirth: 31.5% (45/143) occurred in obstructed or prolonged labour, 15.4% (22/143) in breech birth, with 13 of them having laboured at home, some with histories of prolonged second stage, with head entrapment or arrival in the clinic with entrapment of the head after trial at home.

The multivariate model shows a strong association of APH (adjOR 29.50 (95%CI 21.59-39.84), followed by pre-eclampsia (adjOR 3.53 (95%CI 2.31-5.18) with antepartum stillbirth (Table 5). This is in contrast with intrapartum where APH remained strong but with a slightly lower adjusted odds (adjOR 15.12 (95% CI 7.39-28.12), followed by eclampsia (adjOR 6.76 (95% CI 1.10-21.87). Similar adjusted odds ratios (adjOR) of both antepartum and intrapartum stillbirth (respectively) were observed for small for gestational age (3.14 (95% CI 2.47-3.98) and 2.26 (95%CI 1.42-3.55)) and hypertension (1.78 (95% CI 1.24-2.49) and 2.09 (95% CI 1.10-3.64)). Interestingly, literacy was associated with a lower risk of intrapartum stillbirth (adjOR 0.50 (95% CI 0.29-0.84)), which is not the case for antepartum stillbirth (adjOR 0.95 (95% CI 0.70-1.29)). Characteristics of the antepartum and intrapartum stillbirths are given in Table D in S1 Appendix.

## Discussion

In an unstable area, with intermittent conflict and displacement such as the border of Myanmar and Thailand where marginalised groups including refugee and migrant women experience healthcare access issues, it was possible to reduce stillbirths more than three-fold. This reduction took place in the first two decades (between 1986 and 2010) but since then the rate has stalled. This reduction has been done by opening clinics first for refugees and later for migrants, hereby increasing access to quality healthcare with skilled birth attendants. Together with opening the clinics and actively screening and treating the population for malaria, the number of malaria infections, *especially P. falciparum* has decreased, which contributes to the decrease in stillbirths in the same time period. The plateau has occurred despite an increase in the proportion of births taking place in referral hospitals. This may be partly explained by delays occurring at different levels of care, including presentation to the first health facility (which may be exacerbated by crises as COVID-19 and armed conflict), referral processes and delays in receiving a timely caesarean section at the referral hospital.

**Table 2. Maternal characteristics.**

| | | Livebirth (n = 64,380) | Stillbirth (n = 721) |
|---|---|---|---|
| Mean^ Age | *years* | 26.1 (6.6) [13-53] | 28.3 (7.5) [15-45] |
| Age group | < 18 years | 6.1 (3,929/64,351) | 4.9 (35/720) |
| | 18-34 years | 80.3 (51,670/64,351) | 70.5 (507/720) |
| | ≥ 35 years | 13.6 (8,752/64,351) | 24.7 (178/720) |
| Mean^ Parity | | 2 (2) [0-17] | 2 (2) [0-13] |
| Parity group | 0 | 33.1 (21,249/64,273) | 31.0 (222/717) |
| | 1-4 | 57.6 (36,997/64,273) | 49.3 (355/717) |
| | >4 | 9.4 (6,027/64,273) | 19.5 (140/717) |
| Anaemic | | 35.1 (20,165/57,519) | 43.1 (259/601) |
| Smoking | | 19.4 (42,902/53,197) | 31.2 (165/529) |
| ANC visit in 1st trimester | | 42.6 (27,448/64,380) | 35.5 (256/721) |
| Mean^ 1st trimester BMI | | 21.2 (3.2) [13.5-45.1] | 21.5 (3.6) [15.7-34.3] |
| Asian BMI group | <18.5 | 17.3 (3,350/19,348) | 16.6 (26/157) |
| | ≥18.5-<23 | 59.3 (11,479/19,348) | 58.0 (91/157) |
| | ≥23-<25 | 11.9 (2,297/19,348) | 10.2 (16/157) |
| | ≥25 | 11.5 (2,222/19,348) | 15.3 (24/157) |
| Literacy | | 65.2 (20,437/31,331) | 56.8 (158/278) |
| Status | Refugee | 55.0 (35,407/64,380) | 63.1 (455/721) |
| | Migrant | 45.0 (28,973/64,380) | 36.9 (266/721) |
| Number of ANC visits | ≥4 | 88.2 (56,762/64,374) | 76.4 (549/719) |
| | ≥8 | 69.4 (44,675/64,374) | 55.2 (397/719) |
| Malaria in pregnancy | *P. vivax* | 5.9 (3,777/64,380) | 4.9 (35/721) |
| | *P. falciparum* | 6.8 (4,394/64,380) | 12.5 (90/721) |
| Syphilis | | 0.6 (137/23,441) | 1.5 (3/201) |
| HIV | | 0.5 (118/26,137) | 2.3 (5/221) |
| Antepartum haemorrhage | | 0.7 (391/54,986) | 16.7 (97/582) |
| Hypertension in pregnancy | | 5.0 (3,206/64,380) | 8.7 (63/721) |
| Pre-eclampsia | | 1.8 (1,129/64,380) | 4.4 (32/721) |
| Eclampsia | | 0.2 (149/64,380) | 0.4 (3/721) |
| Mean^ EGA birth, weeks | | 39 (2) (28-45] | 36 (4) [28 27–4144] |
| Preterm birth group, weeks | 28 + 0 to <32 | 0.8 (514/64,380) | 22.1 (159/721) |
| | 32 + 0 to <37 | 7.9 (5,062/64,380) | 32.2 (232/721) |
| | Term ≥ 37 | 91.3 (58,804/64,380) | 45.8 (330/721) |
| Congenital abnormality | | 1.5 (966/63,807) | 14.6 (92/631) |
| Small for gestational age | | 23.4 (13,032/55,808) | 45.7 (208/455) |
| Large for gestational age | | 3.0 (1,696/55,808) | 4.4 (20/455) |
| Mean^ Birthweight g | | 2,954 (468) [600-5580] | 2,054 (842) [520-4100] |
| Place of birth | SMRU | 66.1 (38,658/58,473) | 53.7 (330/614) |
| | Home | 23.2 (13,573/58,473) | 16.0 (98/614) |
| | Hospital | 9.5 (5,544/58,473) | 28.8 (177/614) |
| | Other | 1.2 (698/58,473) | 1.5 (9/614) |
| Caesarean section | | 4.6 (2,964/64,380) | 9.0 (65/721) |
| Type of vaginal birth: | Vertex | 96.8 (59,448/61,416) | 82.9 (544/656) |
| | Breech | 1.3 (807/61,416) | 13.0 (85/656) |
| | Other | 0.2 (101/61,416) | 0.3 (2/656) |
| | Instrumental | 1.6 (966/61,416) | 3.1 (20/656) |

Data are % (n/N) unless otherwise stated, Abbreviations: e.g., ANC antenatal care, EGA estimated gestational age, g grams, N.A. not applicable,

^Mean values consistently expressed as *Mean (sd) [min-max]*

**Table 3. Multivariate regression analysis for factors associated with stillbirths.**

| | | Adj Odds (95% CI) |
|---|---|---|
| Age, years[a] | < 18 | 0.89 (0.59-1.20) |
| | 18-34 | Reference |
| | ≥ 35 | 1.84 (1.49-2.26) |
| Parity[b] | | |
| 0 | 0 | 1.26 (1.05-1.51) |
| 1-4 | 1-4 | Reference |
| >4 | >4 | 1.65 (1.31-2.07) |
| Literacy[c] | | 0.82 (0.64-1.06) |
| Smoking[c] | | 1.47 (1.19-1.80) |
| HIV[c] | | 5.06 (1.77-11.39) |
| Syphilis[c] | | 1.66 (0.27-5.31) |
| Congenital abnormality[c] | | 11.17 (8.79-14.05) |
| Malaria[d] | P. vivax | 0.77 (0.49-1.14) |
| | P. falciparum | 1.62 (1.16-2.20) |
| Hypertensive diseases[d] | Hypertension in pregnancy | 1.83 (1.36-2.42) |
| | Pre-Eclampsia | 2.78 (1.88-3.96) |
| | Eclampsia | 1.55 (0.26-4.92) |
| Antepartum haemorrhage[e] | | 24.28 (18.28-31.94) |
| Anaemia[d] | | 1.10 (0.90-1.35) |
| Small for gestational age[e] | | 3.01 (2.45-3.70) |
| Large for gestational age[e] | | 1.42 (0.70-2.55) |

Corrected for: [a] parity and year of birth, [b] age and year of birth, [c] age, parity and year of birth, [d] age, parity, smoking, P.vivax, P. falciparum, and year of birth, [e] age, parity, smoking, P. vivax, P. falciparum, Hypertensive diseases (Pre-Eclampsia, Hypertension in pregnancy, Eclampsia) and year of birth. There is no correction for literacy, HIV, birthweight category due to high missing values.

The overall stillbirth rate of 11 per 1000 births, is similar in comparison to other low-resource settings in the region, Myanmar 12.8 and Laos 14.3 per 1000 births but higher than Thailand with 3.9 per 1000 births in 2023 [42]. It is expected that the stillbirth rate is increased relative to the local Thailand population when access to all levels of health care for migrant and refugee women is not as equitable as for the local population [10]. Further strengthening the relations between facilities with basic emergency obstetric care [27] and hospital care to improve referral and risk management has the potential to improve the outcomes further [28].

It was possible to retrospectively classify the majority of the stillbirths over three decades. The most important cause of stillbirth in this population was APH particularly due to placental abruption. This is similar to a study in Suriname where it explained 26% of causes of stillbirths but in other low resource setting it ranges from 0.1-92% [5,29]. Even in high resource settings it can be a cause of up to 20% of perinatal deaths [30]. Heat stress and air pollution increase the risk of placental abruption and could possibly increase the risk of stillbirth in this population, as migrant women work in hot, hard conditions, (e.g., the agricultural or construction sectors) and reside in sub-optimal housing [31–35]. Smoking rates have significantly decreased over time, with a small decrease in APH, which may represent the ongoing impact of indoor (cooking or partner) smoke exposure dampening the beneficial effects of lifestyle changes. Improving living conditions of this vulnerable population and referral management could decrease the mortality from APH and in particular placental

**Table 4. Causes of stillbirth antepartum and intrapartum.**

| Antepartum (n = 425) | | Intrapartum (n = 149) | |
|---|---|---|---|
| **Causes** | **N (%) (total n = 425)** | **Causes** | **N (%) (total n = 149)** |
| Unknown | 117 (27.5) | Unknown | 6 (4.1) |
| **KNOWN CAUSES** | **N = 308** | | **N = 143** |
| Antepartum haemorrhage | 111 (36.0) | Obstructed or prolonged labour | 45 (31.5) |
| Maternal infection | 48 (15.6) | Cord | 18 (12.6) |
| Placenta insufficiency | 40 (13.0) | Antepartum haemorrhage | 16 (11.2) |
| Congenital abnormality | 36 (11.7) | Congenital abnormality | 14 (9.8) |
| Hypertensive diseases | 31 (10.1) | Uterine rupture | 13 (9.1) |
| Trauma | 15 (4.9) | Acute intrapartum moment | 11 (7.7) |
| Cord | 12 (3.9) | Trauma | 9 (6.3) |
| Maternal illness | 11 (3.6) | Maternal infection | 7 (4.9) |
| Other | 4 (1.3) | Hypertensive diseases | 4 (2.8) |
| | | Placenta insufficiency | 3 (2.1) |
| | | Other | 3 (2.1) |

**Table 5. Multivariate regression analysis for factors associated with antepartum and intrapartum stillbirths.**

| | | Antepartum Adj Odds (95% CI) | Intrapartum Adj Odds (95% CI) |
|---|---|---|---|
| Age[a] | < 18 | 0.85 (0.49-1.38) | 0.71 (0.31-1.40) |
| | 18-34 | Reference | Reference |
| | ≥ 35 | 1.99 (1.54-2.57) | 1.60 (0.96-2.60) |
| Parity[b] | 0 | 0.96 (0.74-1.23) | 1.72 (1.18-2.48) |
| | 1-4 | Reference | Reference |
| | >4 | 1.85 (1.39-2.44) | 1.29 (0.72-2.24) |
| Literacy[c] | | 0.95 (0.70-1.29) | 0.50 (0.29-0.84) |
| Smoking[c] | | 1.40 (1.09-1.79) | 1.34 (0.85-2.08) |
| HIV[c] | | 4.72 (1.15-12.84) | 3.90 (0.22-18.42) |
| Syphilis[c] | | 2.29 (0.37-7.38) | NA |
| Congenital abnormality[c] | | 9.91 (7.33-13.17) | 12.67 (7.64-20.02) |
| Malaria[d] | P. vivax | 0.67 (0.37-1.10) | 0.57 (0.17-1.37) |
| | P. falciparum | 1.64 (1.10-2.38) | 1.84 (0.94-3.32) |
| Hypertensive diseases[d] | Hypertension in pregnancy | 1.78 (1.24-2.49) | 2.09 (1.10-3.64) |
| | Pre-Eclampsia | 3.53 (2.31-5.18) | 1.62 (0.49-3.88) |
| | Eclampsia | NA | 6.76 (1.10-21.87) |
| Antepartum haemorrhage[e] | | 29.50 (21.59-39.84) | 15.12 (7.39-28.12) |
| Anaemia[d] | | 1.21 (0.95-1.54) | 1.15 (0.74-1.76) |
| Small for gestational age[e] | | 3.14 (2.47-3.98) | 2.26 (1.42-3.55) |
| Large for gestational age[e] | | 1.11 (0.43-2.31) | 2.90 (0.87-7.23) |

Abbreviations: N.A. not applicable

Corrected for: [a] parity and year of birth, [b] age and year of birth, [c] age, parity and year of birth, [d] age, parity, smoking, *P. vivax, P. falciparum,* and year of birth, [e] age, parity, smoking, *P. vivax, P. falciparum,* Hypertensive diseases (Pre-Eclampsia, Hypertension in pregnancy, Eclampsia) and year of birth. There is no correction for literacy, HIV, birthweight category due to high missing values.

abruption [36,37]. The delay of women arriving to the clinic, compounded by delay from the clinic to the hospital, and delay from hospital arrival to the operation room where a caesarean section can be performed is also likely to contribute to this high association with APH and stillbirth and mechanisms to reduce delays need to be considered.

Congenital abnormality was a cause in 8% of the stillbirths and this is in agreement with previously published studies reporting up to 7% [38]. In this population there is ultrasound screening for gestational age assessment and major congenital abnormalities are sometimes detected by the local sonographers. Congenital abnormality scanning could be introduced with training of the local staff if the additional workload can be supported and abnormal findings can be handled in the health system with culturally acceptable care for women and staff. Late ANC after 28 weeks' gestation with a major congenital abnormality contributes to the number of stillbirths, likely more so than in high resource settings where terminations for these conditions are available and late ANC attendance is potentially less likely. Nevertheless, this number reflects a true estimation, presumably relevant in other low resource settings [39].

In this population maternal infections remain an important reason for stillbirth and limited diagnostics limit the ability to assign an exact cause of the fever. Chorioamnionitis was suspected if there was fever surrounding birth as a clinical but not laboratory confirmed diagnosis. Two particular infections - *P. vivax and P. falciparum* – were extensively studied, treated and prevented in this population and there has been a significant decrease of malaria in the border area. Previous analysis of SMRU data has found that both falciparum and vivax malaria were associated with antepartum stillbirth, but only when symptomatic [13]. This analysis extends the data by a further 8 years (2016–2023) of births and includes both symptomatic and asymptomatic infections. Typhus (scrub and murine) has been identified as a cause of stillbirth in the area but testing is not routinely available [40].

The local practice of abdominal massage or fundal pressure by the traditional birth attendant (and/or other family members) at home was provided in the history of cases of trauma and uterine rupture. Traditional home birth practices could also simply cause delayed arrival at the clinic. Reasons for homebirth were not available for this cohort but could include financial limitations, distance, lack of transportation, flooding condition impossibility to come to the clinics, bad experience in hospital/clinic or their personal choice and local tradition [41]. Only place of birth, and not place of labour, was available for the cohort and emergencies (e.g., APH and uterine rupture), were recorded as hospital, even though the labour likely occurred at home.

Literacy was associated with a lower risk of intrapartum stillbirth. As previously reported in this population during the COVID-19 pandemic, literate women were less associated with loss to follow up [17]. This suggests that literate women are more likely to access to healthcare in childbirth compared to illiterate women. This study did not look into health literacy specifically, but as reported earlier, health literacy, understanding of the medical advice and dangers signs given during the antenatal care and perinatal outcomes are closely related [43].

Refugees and undocumented migrants in this population, and worldwide, experience barriers to accessing health care due to issues with finance, documents, transport, health insurance and language [18,44,45]. These barriers may be decreased in settings where humanitarian initiatives can provide assistance adjusted to local needs [46,47]. Migrant antenatal and childbirth services on the Thailand Myanmar border have been supported by NGOs. [48] These collaborations should be strengthened to increase the access to healthcare for migrants. [48] Another report shows that these collaboration could be complicated by financial support, lack of communication or unequal partnerships and of course, the capacity to absorb care of marginalised populations [46]. In this population, an increase in hospital births over the last decade has not correlated with a reduction in stillbirths and this indicates the complexity.

The findings of this study are supported by from other conflict-affected settings. Similar challenges have been described among marginalized and displaced populations in settings such as Lebanon and Gaza, where conflict and limited access to maternal health services and contribute to adverse perinatal outcomes. [49,50] Armed conflict disrupts health systems, results in displacement, reducing the quality of living conditions and shelter, increasing food insecurity and psychosocial stress among pregnant women. These factors, together with barriers to accessing antenatal and obstetric care, have been associated with adverse pregnancy outcomes, including low birthweight and preterm birth. [51]

Classification helps for systematic assignment of the outcomes and potential interventions. While it is useful for comparison of data between countries, it remains rather general, indicating only the most important cause of stillbirth while the cause of most stillbirths is multifactorial. For example, stillbirth due to placenta abruption draws focus to a lack of acute care in labour while the growth restriction due to pre-eclampsia and malaria during pregnancy that contributed to the abruption draw focus to antenatal prevention.

There are limitations to this retrospective study. Earlier antenatal records had a paucity of information compared to current records and those without fetal heartbeat at the time of active labour or the last time of active fetal movement, could not be classified. This dataset connects registration, antenatal events and birth outcomes. The proportion of women lost to follow-up before birth is expected but is likely to underestimate the true stillbirth rate in this mobile population. Loss to follow-up may not have occurred at random, as women lost to follow-up may have been more vulnerable and at higher risk of adverse outcomes, thereby potentially biasing the results. Further population-based research should aim to capture outcomes across the entire population.

Theoretically placental abruption in this population may also be associated with psychological stress because of migration and war but this was not measured and is a limitation [52]. Nevertheless, this remains the largest cohort in a low-resource setting of marginalized refugee and migrant women which provides useful information for improving future care. It shows that with improving the access to basic quality care with skilled birth attendants, the stillbirth rate can be reduced for this population. Further decrease could be attained by strengthening the relation between the facilities with basic and higher levels of care for further management of high risk cases and improving referral systems. Raising awareness on the dangers of uterine pressure/massage could also reduce preventable stillbirth.

## Conclusion

This analysis provides evidence that stillbirth rates among refugee and migrant women can be reduced, even in the context of intermittent conflict. By increasing access to basic quality care, treating malaria and ensuring skilled birth attendance, preventable stillbirths can be reduced. However the observed plateau in stillbirth suggests that achieving parity with the host population remains very difficult due to persistent differences in access to timely and adequate care all levels of health systems between refugee, migrant, and host communities.

## Supporting information

**S1 Appendix. Supplementary tables A-D.**
(DOCX)

**S1 Fig. Direct acyclic graph.**
(TIF)

**S2 Fig. Perinatal indicators over time.**
(TIF)

## Acknowledgments

We would like to thank all the staff and volunteers but especially local staff who worked for SMRU and provided the free care for the migrant and refugee population.

## Author contributions

**Conceptualization:** Taco Jan Prins.

**Data curation:** Taco Jan Prins, Gwen van der Schaaf, Aung Myat Min, Mary Ellen Gilder, Nay Win Tun, May Mon Mon Theint, Jacher Wiladphaingern, Eh Moo, Kerryn A. Moore, Rose McGready.

**Formal analysis:** Taco Jan Prins, Kerryn A. Moore.

**Supervision:** Kerryn A. Moore, Chaisiri Angkurawaranon, Marcus J Rijken, Michele van Vugt, Francois Nosten, Rose McGready.

**Writing – original draft:** Taco Jan Prins.

**Writing – review & editing:** Taco Jan Prins, Gwen van der Schaaf, Aung Myat Min, Mary Ellen Gilder, Nay Win Tun, May Mon Mon Theint, Jacher Wiladphaingern, Eh Moo, Kerryn A. Moore, Jelle Stekelenburg, Chaisiri Angkurawaranon, Marcus J Rijken, Michele van Vugt, Francois Nosten, Rose McGready.

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
