## [Decision Letter · Decision Letter 0]

2 Mar 2026

PGPH-D-26-00283

Reduction of stillbirth rate in refugee and migrant populations living on the Thailand Myanmar border: a retrospective study 1986-2023

Dear Dr. Taco Jan Prins,

Thank you for submitting your manuscript to PLOS Global Public Health. After careful consideration, we feel that it has merit but does not fully meet PLOS Global Public Health’s publication criteria as it currently stands. Therefore, we invite you to submit a new version with minor changes of the manuscript that addresses the points raised during the review process. If any suggested revisions cannot be accommodated, please explain the reasons for their exclusion.

A letter that responds to each point raised by the editor and reviewer(s) should be prepared and uploaded as a separate file labeled *“Response to Reviewers.”* If any suggested revisions cannot be accommodated, please provide a clear explanation for the reasons behind their exclusion.A marked-up copy of your manuscript that highlights changes made to the original version. You should upload this as a separate file labeled 'Revised Manuscript with Track Changes'.An unmarked version of your revised paper without tracked changes. You should upload this as a separate file labeled 'Manuscript'.

We look forward to receiving your revised manuscript.

Kind regards,

Natia Skhvitaridze, M.D. MBA., Ph.D

Academic Editor

Additional Editor Comments:

**Please** review your reference list to ensure that it is complete and correct. If you have cited papers that have been retracted, please include the rationale for doing so in the manuscript text, or remove these references and replace them with relevant current references. Any changes to the reference list should be mentioned in the rebuttal letter that accompanies your revised manuscript. If you need to cite a retracted article, indicate the article’s retracted status in the References list and also include a citation and full reference for the retraction notice.

Reviewers' comments:

**Comments to the Author**

1. Does this manuscript meet PLOS Global Public Health’s publication criteria? Is the manuscript technically sound, and do the data support the conclusions? The manuscript must describe methodologically and ethically rigorous research with conclusions that are appropriately drawn based on the data presented.

Reviewer #1: Yes

Reviewer #2: Yes

2. Has the statistical analysis been performed appropriately and rigorously?

Reviewer #1: Yes

Reviewer #2: Yes

3. Have the authors made all data underlying the findings in their manuscript fully available (please refer to the Data Availability Statement at the start of the manuscript PDF file)?

Reviewer #1: Yes

Reviewer #2: Yes

4. Is the manuscript presented in an intelligible fashion and written in standard English?

Reviewer #1: Yes

Reviewer #2: Yes

5. Review Comments to the Author

Please use the space provided to explain your answers to the questions above. You may also include additional comments for the author, including concerns about dual publication, research ethics, or publication ethics. 

Reviewer #1:

1. Summary

It was an honour for us to appear among the ones who had to review manuscript titled: « Reduction of stillbirth rate in refugee and migrant populations living on the Thailand Myanmar border: a retrospective study 1986-2023 » written by Taco Jan Prins and collaborators from Chiang Mai University, which cover a period of 37 years as shown in the tittle. After a deep reading we noticed that the main idea was to identify the causes and circumstances of stillbirth during the pregnancy and the birth in this kind of marginalized population and what was the possible paths for reducing the issue. The authors noticed that the main causes were the blooding during the pregnancy between 28 weeks of estimated age and the birth, congenital malformation and the maternal infections(specially the malaria) and other specifical ones followed in there frequency as collected in interval of 4 years in the period of study. Beyond that were social condition of displaced people (refugees and migrants) without health assurance. The good news from the study was in spite of marginalisation and lack of health assurance of refugees and migrants, cases of stillbirth decreased in the decades according the assistance for access to antenatal care services for this people in need. The used method to collect data and setting study are very clear, well shown and current statistics measures used to analyze it.

2. Discussion of specific areas to improvement.

I didn’t notice major issue. I think that they can only improve the perspective for next study. Specially, the relationship between stress of migration.

3. Other points

No comment here.

Reviewer #2:

This is an important study that quantifies the progress and persistent challenges in reducing stillbirths for limited studied marginalized population (refugee and migrant women). Expressing sincere appreciation for the authors for conducting this research. The findings have direct implications for policy and practice in similar humanitarian and low-resource settings.

I find it suitable for publication after a minor revision to address the following points and comments:

- Line 52: the death rate mentioned 16.2 for South Asia, this rate covers both central and South Asia (this should be precisely mentioned or alternatively refer to the 16.7 for South Asia only, reference to the same report “Never forgotten”.

- Line 53: Europe SBR is 3.0 as per the same report mentioned in the previous point.

- Line 60: suggest clarification on and addressing the continuum of care in this point of quality care.

- Line 85: has this electronic shift in all clinics at the same time? Please clarify or explain differences.

- Lines 96 - 97: mentioning either side of the borders while it is mentioned this clinic is located in the Thailand side of the border line (are you referring to IDPs (internally displaced population, be specific)), revise this and make it clear for readers. Suggest adding some background on the main dynamics of the two sides of the border in the introduction or under the setting section of the methods.

- Lines 97 – 100: are the SMRUs in Myanmar or Thailand side of the border, these lines are a bit confusing (please review and ensure clarify how the setup of SMRUs system is at its current status and level from which data was collected/extracted!

- Line 101: In brief, specify the “other ethnic groups” for clarity, neutrality & credit.

- Line 106: It will be valuable to know, how many sites covered in the study and how many records, explain the number of facilities and types (clinics, hospitals) and locations (Thailand/Myanmar??) from which the records were collected by type of records (paper/electronic)

- Lines 153 -156: Revisit and clarify this sentence: Due to unknown timing of death most stillbirths at home had an unknown timing of the stillbirth, unless there was a strong indication for antepartum (e.g. severe growth restriction or maceration) or intrapartum (arrived in the clinic with head entrapment at breech delivery).

- Line 212: Results (Table 1): The presentation of percentages is slightly confusing, suggest reviewing and remove the repetition of total number in “Causes N=721 (%)”.

- Line 228: Table 2 page 13: there are missing data for all variables in the table, suggest adding a footnote to mention this for clarity, for example in the Place of birth the denominator is 58,473, but the total livebirths is 64,380 leading to around 5,900 missing their location, was this because this was not recorded (need to clarify this particular variable but also for other variables in the table).

- Line 275: the term quality healthcare should be used within the population context (migration and refugee), there researchers have not referenced to the quality standards for healthcare in such context and this is important to correlate to this situation.

- Lines 284 – 286: support this statement by a reference.

- Lines 292-296: the authors refer to interesting suggestions on occupational hazards, to strengthen this point, could they add a reference to a study documenting the occupational hazards (e.g., agricultural work in heat) or housing conditions of this specific population, if such data exists?

- Line 305: the authors used “Anomaly” here, suggest using the same terminology across the different sections for consistency, although I would recommend using anomaly/anomalies rather than abnormality/abnormalities, but I leave this to authors decision.

- Lines 312-313: revise the sentence for clarity and consider rephrasing to “limit the ability to assign an exact cause …”

- Lines 339-341: please support this statement by reference(s).

- Lines 343-345: is this a finding from this study if so, make it clearer or please support this statement by reference(s).

- Lines 463 – 465 Reference #18 and Line 540-542 Reference #43 seems to be identical and need consolidation, please review and correct.

Additional comments:

1. Discussion of the Plateau 2010-2023, the observation that the stillbirth rate has plateaued since approximately 2010, despite increased hospital births, is a critical finding. The discussion touches on this (lines 277-286, 343-345) but could be strengthened by a more in-depth exploration of potential reasons. Could this be due to the harder-to-reach women who have more complex risk factors? Limitations in the quality of care within referral hospitals (e.g., delays in emergency C-sections, different clinical protocols)? Persistent barriers at the "third delay" at facility, the impact of increasing political instability in Myanmar, especially post-2021, on the health and stress levels of the population? Expanding on these possibilities would add significant value to the discussion.

2. In brief discussion of similar context and findings from other humanitarian settings and conflict affected countries covering similar marginalized groups would be great additions and strengthen the manuscript.

3. Missing Data and impact on findings: The issue of missing data is acknowledged (line 122, S1 Table) and in the footnotes of Table 3. While the authors have handled this by noting adjustments, the potential for bias remains. Could the authors comment briefly in the discussion on the potential direction of bias this might introduce? For example, is it likely that women with missing literacy data are more or less vulnerable? Discussion of this limitation would strengthen the manuscript.

4. The conclusion "Improving access to health clinics and skilled attendance at birth has the potential to reduce preventable stillbirths" is correct, at the same time the paper also shows that after improving access, the rate plateaued. This might need slightly more elaboration on higher-level health system barriers and harmful community practices.

In conclusion, this is a well-conducted and important study. The revisions requested are minor and aimed at enhancing the clarity, depth, and impact of the manuscript. The language, while generally clear, contains a number of minor errors that should be corrected before final acceptance. The data availability statement is appropriate for such vulnerable population. I am confident that the authors can address these points, and I recommend the manuscript for acceptance pending these minor revisions.

6. PLOS authors have the option to publish the peer review history of their article (what does this mean?). If published, this will include your full peer review and any attached files.

**Do you want your identity to be public for this peer review?** For information about this choice, including consent withdrawal, please see our Privacy Policy.

Reviewer #1: **Yes:** Roger Paluku Hamuli

Reviewer #2: No

Figure Resubmissions:

---

## [Decision Letter · Decision Letter 1]

14 Apr 2026

Reduction of stillbirth rate in refugee and migrant populations living on the Thailand Myanmar border: a retrospective study 1986-2023

PGPH-D-26-00283R1

Dear Msc Prins,

We are pleased to inform you that your manuscript 'Reduction of stillbirth rate in refugee and migrant populations living on the Thailand Myanmar border: a retrospective study 1986-2023' has been provisionally accepted for publication in PLOS Global Public Health.

Best regards,

Julia Robinson

Executive Editor

Reviewer Comments (if any, and for reference):

Reviewer's Responses to Questions

**Comments to the Author**

1. If the authors have adequately addressed your comments raised in a previous round of review and you feel that this manuscript is now acceptable for publication, you may indicate that here to bypass the “Comments to the Author” section, enter your conflict of interest statement in the “Confidential to Editor” section, and submit your "Accept" recommendation.

Reviewer #1: All comments have been addressed

Reviewer #2: All comments have been addressed

2. Does this manuscript meet PLOS Global Public Health’s publication criteria? Is the manuscript technically sound, and do the data support the conclusions? The manuscript must describe methodologically and ethically rigorous research with conclusions that are appropriately drawn based on the data presented.

Reviewer #1: Yes

Reviewer #2: Yes

3. Has the statistical analysis been performed appropriately and rigorously?

Reviewer #1: Yes

Reviewer #2: Yes

4. Have the authors made all data underlying the findings in their manuscript fully available (please refer to the Data Availability Statement at the start of the manuscript PDF file)?

Reviewer #1: Yes

Reviewer #2: Yes

5. Is the manuscript presented in an intelligible fashion and written in standard English?

Reviewer #1: Yes

Reviewer #2: Yes

6. Review Comments to the Author

Reviewer #1: In this case, I don't have a comment to make and I recommend if possible the publication after the last hand

Reviewer #2: My recommendation is to accept this version and appreciate the addressing of all comments. Minor grammatical correction in 2 places as following:

- Line 86 delete “in” before “from”.

- Line 96 add the word “of” before the “Thai-Myanmar border… “

7. PLOS authors have the option to publish the peer review history of their article (what does this mean?). If published, this will include your full peer review and any attached files.

**Do you want your identity to be public for this peer review?** For information about this choice, including consent withdrawal, please see our Privacy Policy.

Reviewer #1: **Yes:** PALUKU HAMULI Roger

Reviewer #2: **Yes:** Dr. Rashad Sheikh
